# Protection against Paraquat-Induced Oxidative Stress by *Curcuma longa* Extract-Loaded Polymeric Nanoparticles in Zebrafish Embryos

**DOI:** 10.3390/polym14183773

**Published:** 2022-09-09

**Authors:** Ana Teixeira, Marisa P. Sárria, Inês Pinto, Begoña Espiña, Andreia C. Gomes, Alberto C. P. Dias

**Affiliations:** 1Centre for the Research and Technology of Agro-Environment and Biological Sciences (CITAB), Department of Biology, University of Minho, Campus of Gualtar, 4710-057 Braga, Portugal; 2CBMA (Centre of Molecular and Environmental Biology), Department of Biology, University of Minho, Campus of Gualtar, 4710-057 Braga, Portugal; 3INL—International Iberian Nanotechnology Laboratory, Avenida Mestre José Veiga, 4715-330 Braga, Portugal; 4IB-S—Institute of Science and Innovation for Sustainability, University of Minho, Campus of Gualtar, 4710-057 Braga, Portugal

**Keywords:** curcumin, zebrafish embryogenesis, oxidative stress, neuroprotection, PCL nanoparticles, paraquat

## Abstract

The link between oxidative stress and environmental factors plays an important role in chronic degenerative diseases; therefore, exogenous antioxidants could be an effective alternative to combat disease progression and/or most significant symptoms. *Curcuma longa* L. (CL), commonly known as turmeric, is mostly composed of curcumin, a multivalent molecule described as having antioxidant, anti-inflammatory and neuroprotective properties. Poor chemical stability and low oral bioavailability and, consequently, poor absorption, rapid metabolism, and limited tissue distribution are major restrictions to its applicability. The advent of nanotechnology, by combining nanosacale with multi-functionality and bioavailability improvement, offers an opportunity to overcome these limitations. Therefore, in this work, poly-Ɛ-caprolactone (PCL) nanoparticles were developed to incorporate the methanolic extract of CL, and their bioactivity was assessed in comparison to free or encapsulated curcumin. Their toxicity was evaluated using zebrafish embryos by applying the Fish Embryo Acute Toxicity test, following recommended OECD guidelines. The protective effect against paraquat-induced oxidative damage of CL extract, free or encapsulated in PCL nanoparticles, was evaluated. This herbicide is known to cause oxidative damage and greatly affect neuromotor functions. The overall results indicate that CL-loaded PCL nanoparticles have an interesting protective capacity against paraquat-induced damage, particularly in neuromuscular development that goes well beyond that of CL extract itself and other known antioxidants.

## 1. Introduction

The world is experiencing a dramatic increase in age-related diseases like cancer, cardiovascular and neurodegenerative disorders. Besides genetic predisposition, there is significant evidence of the link between oxidative stress and environmental factors with incidence towards chronic pathologies [1,2,3,4]. In this regard, the development and progression of some of these pathologies like Parkinson and Alzheimer’s disease have been linked to the exposure of environmental chemicals, such as herbicides [5,6,7]. These agrochemicals may induce overproduction of free radicals, mitochondrial dysfunction and damage of the antioxidant system, resulting in impaired cell functioning and production of more toxic species [6,7,8].

Regulating the production and elimination of these reactive species is the key for modulation of critical cellular functions [2,9,10]. Accordingly, it is suggested that the use of exogenous antioxidants could be an effective alternative to combat the progression of major human degenerative diseases [11,12,13]. Some studies have shown that medicinal plants are an excellent source of molecules with incomparable chemical diversity and high bioactivity, which can be effective preventives of various chronic diseases [12,14,15].

*Curcuma longa* L. (CL), commonly known as turmeric, is a popular food condiment that stands out due to a wide range of bioactivities. Turmeric contains a variety of interesting phytochemicals as curcumin (CC), which is assigned to be the most active agent [16,17]. This yellow-colored compound is a multivalent molecule described as having antioxidant activity [18,19], anti-inflammatory [20], anti-cancer [21], cardioprotective [22], neuroprotective [23,24,25] and hepatoprotective properties [26,27]. CC was also proven to be a safe agent for in vivo application [28]. However, the clinical use of this polyphenolic constituent is limited due to its poor chemical stability and low oral bioavailability and, consequently, poor absorption, rapid metabolism, and limited tissue distribution [29,30]. Due to its chemical instability, at physiological pH, CC rapidly degrades to bicyclopentadione by autoxidation. Furthermore, due to the presence of metabolizing enzymes, CC may undergo biotransformation to form reduction products and glucuronide in the liver and other organs [31]. As a result, low serum levels are quantified. The work [32] has shown that after CC oral administration to healthy volunteers, minimal quantities were quantified in serum level (in 10 g only 50.5 ng/mL has quantified). After absorption, CC is subjected to conjugation at various tissue sites, the liver being indicated as the major organ responsible for its metabolism [33]. Additionally, the systemic elimination from the organism is another contributing factor that affects CC bioavailability and consequently interferes with its biological characteristics [33]. Therefore, a formulation with higher solubility and controlled release property would be advantageous for the therapeutic application of CC.

Nanotechnological approaches, such as the development of an efficient drug delivery system [34,35,36,37,38,39,40], may help overcome these limitations of CC-based therapies by combining nanoscale with multi-functionality and bioavailability improvement of active molecules [24,30,41]. Polymeric nanoparticles, liposomes, microemulsions, and solid lipid nanoparticles have been designed and synthetized in order to do so [24,25,42]. These nanocarriers improve the solubility and stability of active compounds, prevent premature degradation, enhance uptake, control drug release, and target the formulation to specific tissues or organs [42]. Despite extensive research into drug delivery vehicles, new materials are being continuously developed without information about their long-term toxicological impact. Thus, prior to application as nanotherapeutics, their toxicity and bioavailability validation are required [43].

Animal models have recognized importance in the study of pathogenesis and therapeutic strategies of human diseases. Among small vertebrate species, zebrafish (*Danio rerio* Hamilton, 1822) has achieved high popularity in the last years for modeling human diseases and disorders [44,45,46]. This organism exhibits many organs and cell types similar to that of mammals, and most genes have significant homology with humans [44]. These characteristics, together with its low cost of maintenance, ease of husbandry, high fecundity, rapid development, external fertilization and development, and optical transparency of the embryos have made it a popular model in various fields of research, such as neuroscience, cardiovascular studies, genetics and in (eco)toxicology (e.g., [24,47,48,49,50]). The zebrafish embryotoxicity test (known as Fish Embryo Acute Toxicity test, FET) is a recommended OECD protocol for the assessment of the toxic effects from exposure to environmental chemicals to nano-/micro-sized particles [24,43,50,51,52]. This method offers the complexity of a whole vertebrate system and provides the possibility to study both acute and chronic toxicity effects at every moment of the early developmental window.

Taking into consideration the therapeutic potential assigned to CL and to CC, this work was focused on bioactivity assessment of the methanolic extract of this plant (i.e., CL), free or encapsulated in poly-Ɛ-caprolactone (PCL) nanoparticles, against paraquat-induced oxidative damage. Herbicide paraquat is known to cause oxidative damage and greatly affect neuromotor functions [6,8,53]. The lower cost, high permeability, compatibility with a vast range of drugs, long-term degradation, non-toxic nature, and compatibility with several tissues are some advantages that make PCL, and the chosen polymer, suitable to compose an effective delivery system [25,54,55]. Toxicity assessment of the developed nanosystems was investigated using zebrafish embryos, following OECD FET test guideline 236 [56].

## 2. Materials and Methods

### 2.1. Chemicals and Reagents

Paraquat (PQ; CAS number 75365-73-0), gallic acid (GA), curcumin (CC) from *Curcuma longa* L. (turmeric), dimethyl sulfoxide (DMSO), methanol, and poly-Ɛ-caprolactone (PCL) were purchased from Sigma-Aldrich, Lisbon, Portugal. Acetone was obtained from Merck, Lisbon, Portugal; Pluronic^®^ F-68 from AppliChem, Lisbon, Portugal and formic acid from VWR, Lisbon, Portugal.

### 2.2. Compounds and Nanoparticles

PQ, GA, CL, and CL-PCL NPs were defined as experimental conditions. All tested solutions and suspensions were prepared in DMSO and then diluted using ultrapure water (18.2 MΩ cm at 25 °C). To guarantee that an increase in mortality or malformation in zebrafish embryos were not solvent-induced, the final concentration of DMSO considered was below 0.10% (*v*/*v*) [57,58].

### 2.3. Preparation and Characterization of CL Plant Extract

Rhizomes of CL collected from India were dried under shade and finely powdered. A volume of 50 g of biomass was extracted in 100% methanol solution with cycles of sonication at room temperature (RT), in the dark. Following filtration, the extract was concentrated using a rotary evaporator and stored at −80 °C. After lyophilization, CL was kept at RT in the dark and protected from moisture. For CL extract characterization, the liquid phase was filtered (Ø 20 µm), and the sample in the proportion 1:10 (extract: methanol) was analyzed by HPLC-DAD (HITACHI, LabChrom Elite, Tokyo, Japan) and monitored by the computer software EZChrome elite (Agilent Technologies, v 3.02, Santa Clara, CA, USA). The compounds separation was performed on a reversed phase LiChroCART 250-4 column (Phosursohere. RP-18e. 5 μm, Merck, Darmstadt, Germany) at RT, using acetonitrile containing 0.1% (*v*/*v*) formic acid (ACN–FA) and ultrapure water containing formic acid 0.1% (*v*/*v*) (upW-FA), as the mobile phases. The flow rate was 0.8 mL/min, and the elution gradient was 5% (*v*/*v*) of ACN-FA at time 0 min, 30% (*v*/*v*) of ACN-FA at time 30 min, 90% (*v*/*v*) of ACN-FA at time 40 min, and 95% (*v*/*v*) of upW-FA at time 60 min. Spectral data from all compounds were accumulated in the range of 230–550 nm, and chromatograms were recorded at 400 nm. CC (and other curcuminoids) were quantified at 400 nm by the external method, using a commercial curcumin standard (>65% purity).

### 2.4. Preparation and Characterization of Curcuma Longa-Polycaprolactone Nanoparticles

CL encapsulated PCL NPs were prepared by solvent displacement method, with a ratio of 1:10. The corresponding mass of PCL and CL were dissolved in 4 mL of acetone and transferred drop by drop into an aqueous solution containing 1% (*v*/*v*) of Pluronic^®^ F-68, under continuous magnetic stirring (450 rpm) at RT, for 4 h. The NPs suspension was centrifuged (12,000 rpm for 30 min at 4 °C; Eppendorf centrifuge 5804 R, Hamburg, Germany), lyophilized, and stored in a moisture free environment for further use. Empty-PCL NPs were also prepared in similar conditions with an absence of the CL extract in its composition. The particle size and polydispersity index (PdI) were determined by dynamic light scattering (DLS), and the surface charge was determined by zeta potential using Zetasizer Nano ZS (Malvern Panalytical, Malvern, UK). Results are the mean of five test runs. All data were expressed as means ± standard deviation (SD).

### 2.5. Toxicity Evaluation through Zebrafish Model

#### 2.5.1. Zebrafish Husbandry and Maintenance

Adult wild-type zebrafish was maintained at 26.0 ± 1.0 °C in a recirculating aquaria system under a photoperiod of 14:10 h (light:dark) at the zebrafish facility of the International Iberian Nanotechnology Laboratory (INL, Braga, Portugal), to be used as breeding stock. Spawners were screened for diseases and had never been previously exposed to a chemical insult. The animals were fed ad libitum twice a day with commercial flakes and supplemented with live brine shrimp eggs. For egg production, zebrafish males and females at a sex ratio of 2:1 were placed into 30 L aquarium coupled with a bottom-open net cage. Spawning was triggered once the light onset in the following morning. Newly fertilized eggs were collected and washed, and viable zygotes were selected for experiments

#### 2.5.2. Zebrafish Embryotoxicity Test

Nanotoxicity and bioactivity of CL (free and loaded in PCL NPs) were assessed using the zebrafish embryotoxicity test (OECD, FET test guideline 236 [56]) as described in [59,60]. During the experimental period, the resulting embryos were kept at 28.0 ± 1.0 °C with 14:10 light:dark cycle. At selected time-points (Table 1), lethal and sub-lethal effects were assessed via morphological and physiological developmental features analysis that are characteristic of zebrafish embryogenesis [61]. FET experiments were considered valid for a mortality rate inferior to 25% in the control group. All experiments were carried out in strict accordance with the Council of Europe, Directive 2010/63/EU (revised Directive 86/609/EEC), on the protection of experimental animals, including the fact that these were carried out up to 80 h post-fertilization (h_pf_).

##### Toxicity Assessment

To study the toxicity of PQ, GA, and CL, two types of FET assays were carried out to assess long and short insults. In the long experiments (Figure 1A), zebrafish embryos were exposed to the test conditions from 2 until 80 h_pf_. For this period, four time-points during embryonic development were considered: 8, 32, 56, and 80 h_pf_. Each experiment was performed with 40 embryos per condition. In the short exposure (Figure 1B), zebrafish embryos with 2, 4, 8, and 24 h_pf_ were exposed for 24 h after which were analyzed in the microscope. Collected data were of 20 embryos per condition. During the experiments, all embryos were constantly checked for mortality. The incubation medium was renewed daily to ensure oxygenation. Mortality was defined as an embryo that lacked cardiac function, blood circulation motility, and/or was in a certain state of degradation. The parameters studied and respective timepoints are listed in Table 1.

##### Bioactivity Assessment

With the goal of evaluating the potential protective effect of CL extract and CL-PCL nanoparticles, post-incubation tests were carried out (Figure 1C). In these experiments, 2 h_pf_ zebrafish embryos were exposed to PQ for 24 h. Zebrafish embryos were kept until 80 h_pf_ with daily medium renovation and successive mortality checks. Each experiment was performed with 10 embryos, in quadruplicates.

### 2.6. Statistical Analysis

Microscopy images were measured and analyzed by Fiji software. Statistical analysis was performed using Statistica software (StatSoft v.8, Tulsa, OK, USA). Prior to data analysis, all assumptions were met, testing for normality (Shapiro–Wilk test) and homogeneity of variances (Levene’s test). Results were presented as mean ± standard deviation (SD) and *p* value < 0.05 was considered as statistically significant. Details on specific statistical analysis are referred in the legend of the figures. All data were plotted by GraphPad Prism software (GraphPad Inc., v.6, San Diego, CA, USA).

## 3. Results

### 3.1. CL Extract and Nanoparticles Characterization

The chemical profile of methanolic CL extract was investigated in order to measure the concentration of total CC and other curcuminoids. The chromatograms of commercial standard curcumin and our CL extract have similar composition (Figure 2). As previously reported [62], based on the spectral data and retention times obtained, the major compounds found were curcumin and, at lower amounts, the curcuminoids bisdemethoxycurcumin and demothoxycurcumin. From the DAD data collected, the CL extract did not show other significant compounds besides these. Our CL extract contains a total of 459 µg/mL of CC and its curcuminoids. For clarity, the HPLC results relating to the period between 38 and 44 min are presented in Appendix A.

The prepared nanoparticles were characterized in terms of size distribution, PdI, and zeta potential. Analyzing Figure 3, notorious differences were observed between empty and CL-PCL nanoparticles. The former presented higher sizes and PdI values (Figure 3A), with significant variations in long-term (one-way ANOVA, *F*(3,14) = 106.240, *p* < 0.001). Furthermore, surface charge of the developed empty nanoparticles showed a significant decrease along time: at day 0, measured zeta potential was −8 mV and, 4 weeks later, −18 mV (Figure 3B) (one-way ANOVA, *F*(3,14) = 409.630, *p* < 0.001). Regarding the CL-PCL nanoparticles, it was observed that 4 weeks after production, the nanoformulation presented values of size and PdI of 279 nm and 0.26, respectively (Figure 3C). Additionally, it presented a highly negative surface charge at weeks 2, 3, and 4 (>20 mV) (Figure 3D). Significant differences were found in the size and PdI value at weeks 2 and 4 (one-way ANOVA, size: *F*(3,12) = 10.890, *p* < 0.01; PdI value: *F*(3,12) = 11.400, *p* < 0.01), and in the zeta potential at week 0 (one-way ANOVA, *F*(3,12) = 57.770, *p* < 0.001).

### 3.2. Bioactivity Validation in Zebrafish

#### 3.2.1. Toxicity Profiling

In the present study, the short and long terms toxicity of PQ, GA, CL, and CL nanoparticles exposure to zebrafish embryos were evaluated. Regarding PQ long exposure results, notorious effects were observed among the independent variables evaluated (Table 2). The longer exposure of zebrafish embryos to 0.64 µg/mL of PQ caused a significant decrease on pupil surface (ANCOVA, *F*(4,41) = 7.882, *p* < 0.001) and a significant decrease on yolk extension (ANCOVA, *F*(4,43) = 0.998, *p* < 0.05). Concerning the neuro-motor coordination indicator parameters, this herbicide negatively affected zebrafish embryos in a concentration dependent manner. A significant decrease in the cardiac frequency was registered (one-away ANOVA, *F*(4,40) = 7.746, *p* < 0.001), and an increase in the percentage of embryos with spontaneous movements (SM) and of larvae exhibiting free-swimming (FS) (SM: chi-square test, χ^2^ = 11.53, *p* < 0.001; FS: chi-square test, χ^2^ = 16.34, *p* < 0.01) were recorded. In comparison with the control group (that is, 0.00 µg/mL of PQ), 0.64 µg/mL of PQ triggered an increase of 20 and 25% in the number of SM and FS, respectively. Moreover, it was observed that increasing PQ (nominal) test concentration caused toxicity in a time-dependent manner. The exposure to the highest (nominal) test concentration of PQ (i.e., 1.28 µg/mL) resulted in a reduction of 55% in embryo survival (Appendix A).

A well-known antioxidant compound—GA [63] was considered to validate its potential to mitigate the oxidative stress. Accordingly, for GA (Table 2), it was observed that the range of the tested (nominal) concentrations did not interfere with the epibolic arc development and embryos straightening. At 32 h_pf_, however, the embryos exposed to 25 µg/mL GA presented a significant reduction of their yolk volume (ANCOVA, *F*(3,25) = 2.113, *p* < 0.05). Furthermore, it was observed that, although the highest (nominal) test concentration of GA (50 µg/mL) did not interfere with the percentage of zebrafish embryos with SM, effects on the cardiac frequency (one-way ANOVA, *F*(3,16) = 9.143, *p* < 0.001) and on the number of larvae having FS (one-way ANOVA, *F*(3,8) = 2.976, *p* < 0.05) were observed. A volume of 50 µg/mL of GA, in comparison to the untreated embryos, triggered a decrease of approximately 17% on the cardiac frequency and, on the other hand, an increase of 18% to 59% on the FS behavior. In addition, it was recorded that an increased concentration of GA resulted in reduced zebrafish embryonic mortality (Appendix A).

Regarding CL long exposure of zebrafish embryos (Table 2), it is possible to infer that 5 µg/mL of CL did not induce a delay on the embryonic development, did not affect the structures rich in lipids as the yolk, eye, and pupil and did not interfere with the SM and cardiac frequency. However, this extract rich in polyphenolic compounds significantly reduced FS ability (one-way ANOVA, *F*(3,8) = 14.21, *p* < 0.01). Moreover, it was observed that increasing CL concentration (tested at 1.5, 2.5, 5, and 10 µg/mL) led to a decrease in mortality. At 80 h_pf_, zebrafish embryos exposed to 5 µg/mL showed 9% cumulative survival (Appendix A).

Additionally, the results obtained for the short exposure of zebrafish embryos to 5 µg/mL of CL and CL-PCL nanoparticles are summarized in Table 3. In general, 5 µg/mL of either CL Free, CL-PCL, or empty nanoparticles did not interfere with the embryos straightening. However, it was observed that the unformulated drug and the polymer used to encapsulate the plant extract caused a significant increase in the yolk volume (CL Free: ANCOVA, 2 h_pf_: *F*(2,59) = 14.38, *p* < 0.001; 8 h_pf_: *F*(2,59) = 4.18, *p* < 0.05; Empty NPs: ANCOVA, 4 h_pf_: *F*(3,75) = 8.59, *p* < 0.001; 8 h_pf_: *F*(3,82) = 5.30, *p* < 0.01). Concerning the results for zebrafish embryonic cardiac frequency, it was observed that 5 µg/mL of CL Free significantly affected cardiac frequency. The non-treated embryos incubated at 8 h_pf_ showed 2.3 beats/s but, when exposed to 5 µg/mL of CL Free, the cardiac frequency decreased to 1.9 beats/s (one-way ANOVA, 2 h_pf_: *F*(2,37) = 10.26, *p* < 0.001; 8 h_pf_: *F*(2,36) = 15.40, *p* < 0.001; 24 h_pf_: *F*(2,36) = 6.238, *p* < 0.01). At 8 and 24 h_pf_, zebrafish embryos treated with CL-PCL nanoparticles did not show signs of cardiotoxicity. Nevertheless, the empty nanoparticles caused a reduction in zebrafish cardiac frequency, being statistically significant for those incubated at 4, 8, and 24 h_pf_ (Table 3). Furthermore, the effects of CL short exposure on the SM of zebrafish embryos were also recorded. CL Free promoted, in a concentration dependent manner, an increase in the percentage of embryos with SM, which was verified to be statistically significant for embryos incubated at 24 h_pf_ with 5 µg/mL (one-way ANOVA, 24 hpf: *F*(2,7) = 5.331, *p* < 0.05). In contrast, zebrafish embryos exposed to an increasingly higher (nominal) test concentration of CL-PCL nanoparticles showed a decrease in SM. At 4 h_pf_, the number of muscular contractions was significantly reduced from 53 to 33% (one-way ANOVA: 4 h_pf_: *F*(3,9) = 8.987, *p* < 0.01). PCL did not affect the muscular contractions of zebrafish embryos. Furthermore, the short exposure of 5 µg/mL of CL extract (free and encapsulated) seems to have a protective effect on zebrafish embryos survival, as a decrease in mortality rate was observed (Appendix A).

#### 3.2.2. Bioactivity Profiling

The putative protective activity of CL, Free and loaded onto PCL nanoparticles, was evaluated by a FET assay using embryos exposed to PQ. According to the results obtained, 5 µg/mL of CL (free and encapsulated), 25 and 50 µg/mL of GA, and 0.64 µg/mL of PQ were the selected (nominal) test concentrations for a post-incubation test (exposure to PQ then incubation with GA). Results obtained are illustrated in Appendix A. The post-incubation of GA potentiated the survival of zebrafish embryos and was effective in minimizing the negative effects of PQ in the zebrafish muscular contraction at 32 h_pf_. However, it was not able to mitigate cardiotoxicity caused by PQ exposure (Appendix A).

Table 4 and Figure 4 illustrate the results related to the medicinal potential of CL extract (free and encapsulated) against PQ toxicity. According to Figure 4A, it was observed that 13% of the embryos exposed to PQ (and non-treated with any other compound) died, while in conditions where embryos were treated with 5 µg/mL of CL Free and CL-PCL, the mortality rate was significantly lower (i.e., 3% and 0%, respectively). Analyzing the zebrafish embryos cardiac frequency (Figure 4B), it was observed that the toxic effect instigated by PQ exposure was quite remarkable. At 32 h_pf_, the heartbeat of the embryos incubated with PQ decreased significantly to 1.18 beat/s, while in the control group, the embryos presented an average of 1.66 beats/s (Figure 4B). CL Free and the formulated drug counteracted the cardiotoxic effect of PQ by increasing the heart beating of the zebrafish embryos for 1.48 and 1.57 beats/s, respectively (Figure 4B). At 56 h_pf_, the cardiac frequency of zebrafish embryos exposed to PQ was still lower (2.48 beats/s), compared to the control group (2.71 beats/s). However, those treated with CL-PCL nanoparticles exhibited an increased cardiac frequency (2.64 beats/s) (Figure 4B). Differences reported were statistically significant at both time points (32 h_pf_: one-way ANOVA, *F*(7,74) = 24.33, *p* < 0.001; 56 h_pf_: one-way ANOVA, *F*(7,69) = 10.11, *p* < 0.001).

Furthermore, at 32 h_pf_, the percentage of zebrafish embryos with SM decreased from 90% at the control group, to 50% for those exposed to PQ. Nevertheless, the treatment with free CL attenuated the toxic effect on muscular contractions by increasing in 15% the number of embryos with SM (Figure 4C). Differences among conditions were demonstrated to be statistically significant (one-way ANOVA, *F*(7,16) = 13.14, *p* < 0.001). At 80 h_pf_, the effect of 24 h exposure of 0.64 µg/mL of PQ was still notorious (Figure 4D). CL Free and CL-PCL showed to be efficient in reverting the effects caused by PQ. The treatment with these substances increased the larvae FS behavior from 39.2% to 48.6% and 51.4%, respectively (Figure 4D). Differences between conditions were statistically significant (one-way ANOVA, *F*(7,24) = 1.350, *p* < 0.001).

Regarding the morphometric parameters (Table 4), none of the tested conditions seemed to interfere with the zebrafish embryonic development at 8 h_pf_, as well as with the pupil and eye development (32 and 56 h_pf_, respectively). In the early stages of development, zebrafish embryos exposed to PQ showed a significant decrease on yolk volume, compared to the control group (ANCOVA, *F*(7,150) = 3.025, *p* < 0.01). However, the formulated drug counteracts the damage caused by increasing the volume of this structure, and with the progression of the embryonic development, differences on yolk volume tended to dissipate. Aside from this, it was also observed that the post-incubation of CL Free and CL-PCL attenuated the production of reactive oxygen species (ROS) (data not shown).

## 4. Discussion

Oxidative stress plays an important role in mitochondrial activity and consequently on neurodegeneration [3,9,10]. To date, the strategies developed for the treatment of disorders related to neuronal degeneration and cognitive deterioration have been ineffective [9,10,64]. Therefore, the search for new candidates with high antioxidant and anti-inflammatory potential that interact with a wide diversity of molecular targets and pathways has been constant [11,13]. The powdered rhizomes of CL have been used for centuries as a medicinal herb with a wide range of biological applications [16,18,22]. This plant and, in particular, its yellow-colored constituent (CC), has been shown to be very effective in the attenuation of the symptoms associated with neurodegenerative disorders [12,23,24,25].

In this work, in order to potentiate CL activity, polymeric nanoparticles were developed by solvent displacement method, which is a simple, economic, and widely employed technique [25,54,65,66]. Biodegradable polymers have provided numerous avenues to improve drug development and, consequently, the therapeutic efficacy of medicinal entities [67]. PCL is an example of a nontoxic, biodegradable, and biocompatible polymer widely used and approved by the U.S. Food and Drug Administration (FDA) [25,54,55,67]. Considering the obtained results, CL-PCL nanoparticles remain stable for at least four weeks after their production. Comparing CL loaded and empty formulations, large differences were observed in terms of size, homogeneity, and charge potential. CL-loaded formulations are smaller, with broad size distribution and highly negative surface charge. The inclusion of the plant extract seems to potentiate the stabilization of nanoparticles, perhaps by the addition of a negative charge or promoting exposure of negative groups. Physicochemical features of nanoparticles also determine how they interact with cells and tissues. The more negative surface charge of loaded nanoparticles confers more stability also over time, as clearly observed, and it does not cause relevant cytotoxicity issues, which would be the case if the nanoparticles presented cationic charge. More homogeneity in size distribution and slightly smaller particles, when loaded with CL, are also advantageous in terms of predictability of cell response and also facilitate the selection of administration route. In a previous study [25], we optimized the PCL nanoformulations to obtain >70% curcumin encapsulation and also included a release profile of PCL loaded nanoparticles. It reveals that there is sustained release and that, after 168 h, approximately 75% of curcumin is released from the nanoparticles. Regarding the toxicological profile, the free and formulated compounds were demonstrated to be generally inert. As reported by [68], the exposure of zebrafish embryos to these conditions did not induce a delay on development and in toxicity at the level of cardiac function, SM and FS behavior.

The brain and heart are the organs with most demanding energetic necessities and are totally dependent upon oxidative phosphorylation to supply the huge amount of ATP required [69]. In contrast, PQ interferes with electrons transference by activity inhibition of the complex I of mitochondrial respiratory chain, resulting in uncontrolled ROS production [6,70]. Furthermore, the ability of ROS to affect both mitochondrial function and nucleic acids results in critical cellular alterations. This chain of events promotes the phosphorylation of proteins intrinsically associated with neurodegeneration [71], and the reduction of cardiac activity [69]. The obtained results demonstrate that the exposure to PQ caused negative effects in the structures and organs related to zebrafish lipid metabolism. Reference [72] reported that PQ exposure on Kunming mice resulted in lipid peroxidation; increase in ROS levels; and damage in the biliary, gastrointestinal, and nervous systems, in addition to lungs, kidneys, and the liver. Our own results (Appendix A) show that a 24 h incubation with PQ leads to an increase in ROS levels on zebrafish embryos that is maintained for another 24 h, while there is a dose-dependent increase upon exposure to H_2_O_2_. Furthermore, our data suggest that PQ affects neuro-motor coordination, such as trunk contractions and pectoral fin movements, and lead to signs of cardiotoxicity, as a reduced cardiac frequency was observed. These results are concordant with other reports [53,73,74]. PQ exposure on zebrafish results in decreased sensorimotor reflexes, spontaneous movements, and distance swum. Furthermore, the PQ-treated group showed anxiety-like behavior by swimming near the walls [73,74]. In *Drosophila melanogaster,* this herbicide caused alternation of motor-related behavioral patterns, including resting tremors, rotation of the body, and postural instability [68]. According to our results, CL improved survival rate of zebrafish embryos and counteracted the reduction in cardiac frequency as well as the negative effects on the neuro-motor parameters. In Appendix A, it is noticeable that free and formulated curcumin can reduce high ROS levels induced by 24 h contact with PQ. This protective effect of CL seems to be associated with the capacity to restore the endogenous antioxidant system by increasing levels of catalase, superoxide dismutase and glutathione, decreased oxidative markers levels, reduced lipid peroxidation, and counteracted cardiac cell damage [18,75]. Comparing the CL-PCL nanoparticles with the CL Free, the nanosystems have shown to be, in most cases, more competent in mitigating the oxidative damage. This effect may be due to the fact that the encapsulation of CL enhances its bioavailability and pharmacokinetics [25,65,76]. Furthermore, CL was more efficient in attenuating the toxic effect of PQ than GA.

## 5. Conclusions

Our data support the notion that CL possesses antioxidant activity, as well as cardio and neuroprotective properties, validating its potential to minimize damages caused by the herbicide PQ. Delivery of this plant extract incorporated in polymeric nanoparticles also conferred protection against PQ-induced toxicity, in FET assays. The embryos treated with free or formulated CL showed a decrease in mortality rate and cardiotoxicity caused by exposure to the tested herbicide, and the heartbeat of zebrafish embryos increased to values close to those of the control group. Additionally, CL free and CL-PCL minimize the toxic effect of PQ on neuro-motor parameters, as the number of embryos with SM and the number of larvae with FS increased with both treatments. Furthermore, the present work corroborates the potential of zebrafish embryos as a model to assess drug discovery and nanotoxicity.

## Figures and Tables

**Figure 1 polymers-14-03773-f001:**
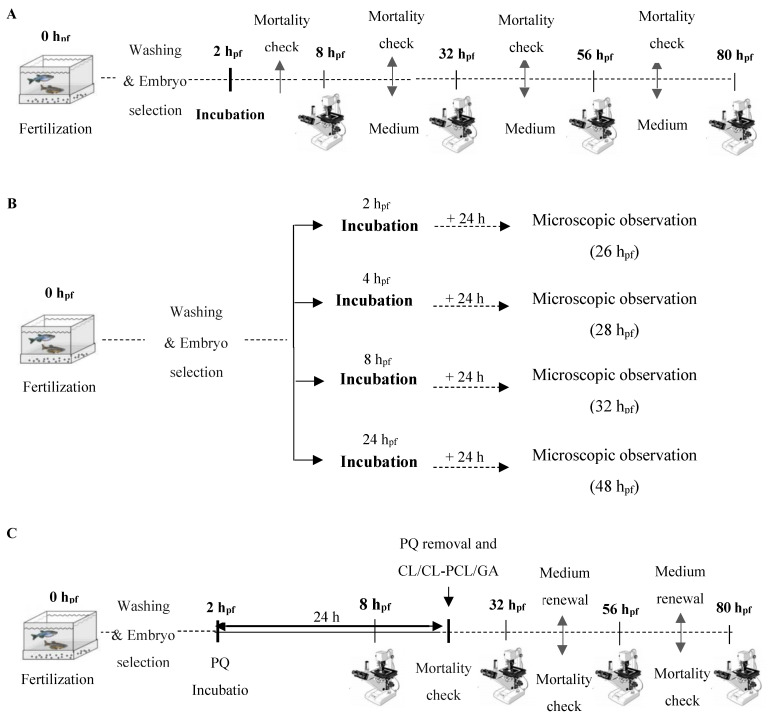
Experimental design of zebrafish embryotoxicity test (FET): (**A**) long exposure; (**B**) short exposure; and (**C**) post-incubation assay. h_pf_—hours post-fertilization; PQ—paraquat; GA—gallic acid (GA), CL—*Curcuma longa* extract (CL) and CL-PCL—CL-poly-Ɛ-caprolactone nanoparticles.

**Figure 2 polymers-14-03773-f002:**
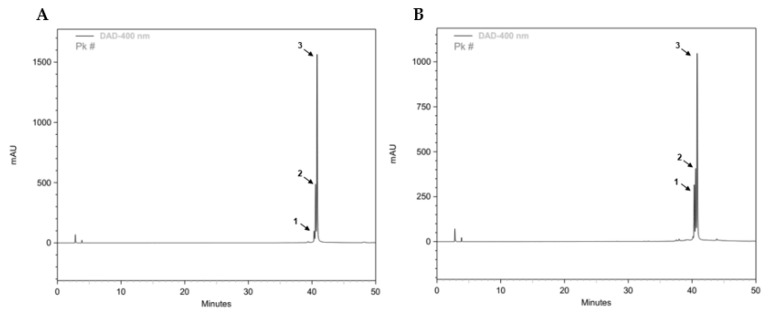
High-performance liquid chromatography (HPLC) analysis of commercial standard curcumin (**A**) and *Curcuma longa* (CL) methanolic extract (**B**). The major compounds present were curcumin (3), demethoxycurcumin (2), and bisdemothoxycurcumin (1).

**Figure 3 polymers-14-03773-f003:**
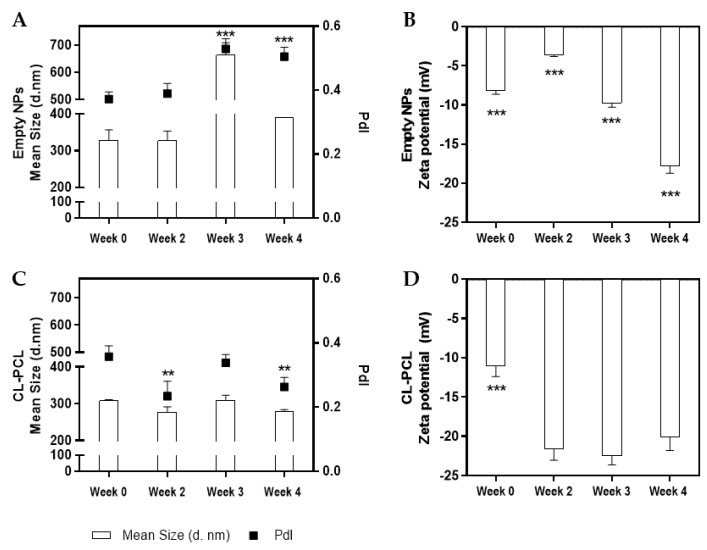
Polymeric nanoparticles characterization: (**A**) mean size (d. nm) and polydispersity index (PdI) of empty nanoparticles (NPs); (**B**) zeta potential of empty NPs; (**C**) mean size (d. nm) and PdI of *Curcuma longa* extract (CL)-poly-Ɛ-caprolactone (CL-PCL) NPs; and (**D**) zeta potential of CL-PCL NPs at different time points. Loaded nanoparticles present values of mean size and PdI lower than the empty ones and are slightly more negatively charged. Both formulations are stable for at least 4 weeks after preparation. Results are expressed as mean ± SD. ** *p* < 0.01; *** *p* < 0.001, one-way ANOVA followed by Student–Newman–Keuls post hoc test.

**Figure 4 polymers-14-03773-f004:**
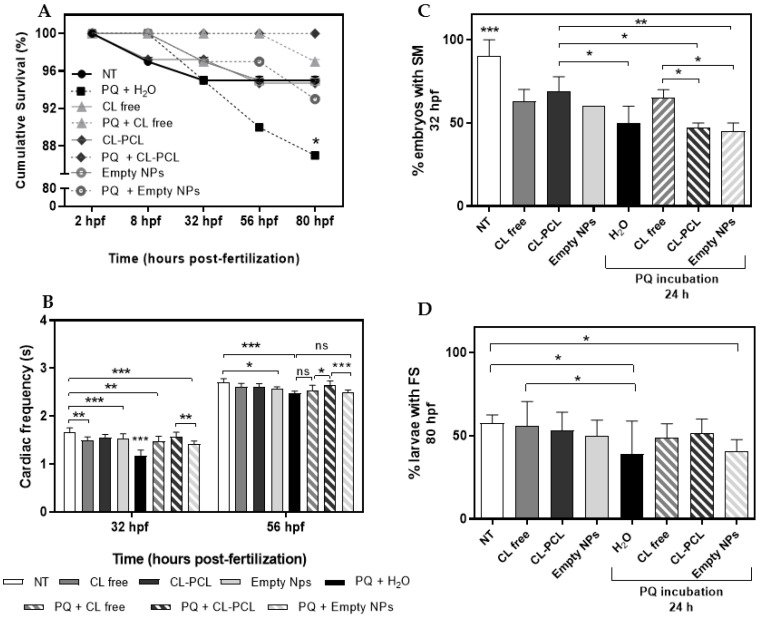
The effects of 24 h insult of paraquat (PQ) followed by a long exposure of *Curcuma longa* extract (CL) Free and encapsulated (CL-PCL) on (**A**) cumulative survival at 8, 32, 56 to and 80 h post-fertilization (h_pf_); (**B**) cardiac frequency (s) at 32 and 56 h_pf_; (**C**) spontaneous movements (SM) at 32 h_pf_, and (**D**) free swimming (FS) at 80 h_pf_ of zebrafish embryos. At 2 h_pf_, the embryos were first exposed for 24 h to 0.64 µg/mL of PQ, followed by an insult of 5 µg/mL of CL free, CL-PCL and empty NPs. CL (free and formulated) reduced the mortality rate. Furthermore, the treatment with CL Free and CL-PCL increased not only the heart frequency of zebrafish embryos exposed to PQ but also increased the number of embryos with SM and the larvae with FS behavior. Results are expressed as mean ± SD. ns—not statistically different, * *p* < 0.05; ** *p* < 0.01; *** *p* < 0.001; one-way ANOVA, followed by Newman–Keuls and Fisher LSD post hoc tests.

**Table 1 polymers-14-03773-t001:** Zebrafish developmental parameters assessed during embryogenesis.

	Time-Point Evaluation (h_pf_—i.e., Hours Post-Fertilization)		
	8 h_pf_	32 h_pf_	56 h_pf_
**Developmental parameters**	Epiboly Yolk volume Egg volume	Head-trunk index Cardiac frequency Yolk volume Egg volume Pupil surface Eye surface Spontaneous movements	Cardiac frequency Yolk volume Yolk extension Hatching

**Table 2 polymers-14-03773-t002:** Zebrafish embryotoxicity test—long exposure toxicity assessment. (+) stands for statistically significant effect and (−) for non-statistically significant effect. h_pf_—hours post-fertilization. CL—*Curcuma longa* extract.

			Paraquat	Gallic Acid		CL
	Independent Variables	h_pf_	0.64 µg/mL	25 µg/mL	50 µg/mL	5 µg/mL
**Morphometric parameters**	Epibolic arc	8	−	−	−	−
	Yolk volume	8	−	−	−	−
	Head-trunk index	32	−	−	−	−
	Yolk volume	32	−	+	−	−
	Pupil surface	32	+	−	−	−
	Yolk volume	56	−	−	−	−
	Eye surface	56	−	−	−	−
	Yolk extension	56	+	−	−	−
**Neuro-motor parameters**	Spontaneous movements	32	+	+	−	−
	Cardiac frequency	56	+	−	+	−
	Free-swimming	80	+	+	+	+

**Table 3 polymers-14-03773-t003:** Zebrafish embryotoxicity test—short exposure toxicity assessment. (+) stands for statistically significant effect and (−) for non-statistically significant effect. h_pf_, hours post-fertilization; CL Free, *Curcuma longa* extract; CL-PCL NPs, *Curcuma longa* extract—poly-Ɛ-caprolactone nanoparticles and empty NPs, empty nanoparticles.

	Independent Variables	h_pf_	CL Free (5 µg/mL)	CL-PCL NPs (5 µg/mL)	Empty NPs (5 µg/mL)
**Morphometric parameters**	Yolk volume	2 4 8 24	+ − + −	− − − −	− + + −
	Head-trunk index	2 4 8 24	− − − −	− − − −	− − − −
	Pupil surface	2 4 8 24	− − − −	− − + +	− − − −
**Neuro-motor parameters**	Cardiac frequency	2 4 8 24	+ − + +	+ + − −	− + + +
	Spontaneous movements	2 4 8 24	− − − +	− + − −	− − − −

**Table 4 polymers-14-03773-t004:** Zebrafish embryotoxicity test—post-incubation experiment. (+) stands for statistically significant effect and (−) for non-statistically significant effect. h_pf_, hours post-fertilization; CL Free—*Curcuma longa* extract (5 µg/mL); CL-PCL NPs—*Curcuma longa* -poly-Ɛ-caprolactone nanoparticles (5 µg/mL) and PQ—paraquat (0.64 µg/mL).

			Post-Incubation Conditions						
	Independent Variables	h_pf_	CL Free	CL-PCL NPs	Empty NPs	PQ ↓ H_2_O	PQ ↓ CL Free	PQ ↓ CL-PCL	PQ ↓ Empty NPs
**Morphometric** **parameters**	Epibolic arc	8	−	−	−	−	−	−	−
	Yolk volume	8	−	−	−	+	+	−	−
	Head-trunk index	32	−	−	−	−	−	−	+
	Yolk volume	32	−	−	−	−	−	−	−
	Pupil surface	32	−	−	−	−	−	−	−
	Yolk volume	56	−	−	−	−	−	−	−
	Eye surface	56	−	−	−	−	−	−	−

## Data Availability

Data will be made available upon reasonable request.

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
