# Peer review of "Protection against Paraquat-Induced Oxidative Stress by Curcuma longa Extract-Loaded Polymeric Nanoparticles in Zebrafish Embryos"

_polymers, 2022, doi:10.3390/polym14183773_

Round 1

Reviewer 1 Report

In this work, the authors used the PCL nanoparticles to incorporate the methanolic extract of Curcuma longa L, evaluated their toxicity in zebrafish embryos, and also analyzed the protective effect on oxidative damage.  The whole paper looks good, but still, needs to be improved before being published. Specific comments list as below.

1. Advanced nanotechnology and developed nanomaterials used in drug delivery, should be included in this paper, like Ding, et al. Materials Today 43 (2021): 166-184, etc. 

2. What the effect of the changes in size and charge of the PCL and functionalized PLC nanoparticles in Figure 3?

3. The mechanism of PCL and CL-PCL nanoparticles reducing the ROS should be discussed. 

4.           Is there any other workable method to evaluate the toxicity of NPs in embryos, like MTT?

5. English still needs to be polished, some used works are not very academic.

Author Response

Reviewer 1

In this work, the authors used the PCL nanoparticles to incorporate the methanolic extract of Curcuma longa L, evaluated their toxicity in zebrafish embryos, and also analyzed the protective effect on oxidative damage.  The whole paper looks good, but still, needs to be improved before being published. Specific comments list as below.

First of all, we acknowledge and appreciate all the relevant comments which we addressed to improve the cohesiveness of the present manuscript.

  1. Advanced nanotechnology and developed nanomaterials used in drug delivery, should be included in this paper, like Ding, et al. Materials Today 43 (2021): 166-184, etc.

Thank you for your suggestion; new works were included in the manuscript.

  1. What the effect of the changes in size and charge of the PCL and functionalized PLC nanoparticles in Figure 3?

The stabilizing effect of the load in the nanoparticles was already addressed in discussion (lines 369-371). More details of the alterations in physicochemical characteristics were discussed from line 371 onward.

  1. The mechanism of PCL and CL-PCL nanoparticles reducing the ROS should be discussed. 

Figures S2 and S3 compile related data and these results are included in the discussion.

  1. Is there any other workable method to evaluate the toxicity of NPs in embryos, like MTT?

Taking into account the assay principle, MTT is enzymatically converted to purple formazan crystals at the reducing microenvironment of metabolically active cells, which are next solubilized using a mixture of organic solvents. Total aqueous-soluble formazan product generated correlates to cellular metabolic activity, and therefore is directly proportional to the number of living cells. Cell toxicity measurements by MTT are therefore dependent of constant cell activity, and signal will be relevant if the cells are exposed to the same amount of MTT. As it is recognized by the vast majority of the most common cytotoxicity assays, a higher variability can arise for missing this assumption, and in case of MTT this probability is relatively higher due to the two-step methodology implied.  In a zebrafish embryo, different tissues and organs are under development, and therefore, cellular activity varies between cells presenting not only different typology, but also different origin. Besides, the use of a mixture of organic solvents (usually 1:1 v/v DMSO:EtOH) to solubilize the colored crystals generated is not compatible to zebrafish embryos viability. Instead, using a detergent for solubilization might be a possibility, but absence of protein precipitation should be ensured. Even at the early cellular stages of zebrafish embryonic development, the determination of metabolically active cells by MTT would be dependent of its permeability towards the chorionic barrier, otherwise tetrazolium salt would not be converted to formazan by active cells. On the other hand, the introduction of a permeabilization step would kill the embryos. Hence, determining toxicity of nanoparticles or nanomaterials in zebrafish embryos by MTT becomes meaningless. Alternatively, toxicity can be quantitatively determined in zebrafish cell lines/embryo derivatives or qualitatively investigated by measuring the heart beat (Sárria et al., 2021). Cardiotoxicity is a known effect derived from nanoparticles and nanomaterials exposure, and in zebrafish embryos there is the advantage of the heart rate being able to be counted through the transparent egg. A correlation can be established between the embryo developmental stage and the heart rate, as compared to a control group at certain hours post-fertilization (see Sárria et al., 2021).

Sárria, M. P., Vieira, A., Lima, Â., Fernandes, S. P., Lopes, I., Gonçalves, A., Gomes, A. C., Salonen, L. M. & Espiña, B. (2021). Acute ecotoxicity assessment of a covalent organic framework. Environmental Science: Nano, 8(6), 1680-1689. DOI: 10.1039/D0EN01059F

  1. English still needs to be polished, some used works are not very academic.

Thank you for the suggestion. We made substitutions and corrections to the language (changes are marked in blue), and added relevant references.

Reviewer 2 Report

The manuscript entitled “Protection against paraquat-induced oxidative stress by Curcuma longa extract-loaded polymeric nanoparticles in zebrafish embryos" shows a novel nanoformulation for effective release and effect of curcuma extract for diverse protective effect against paraquat induced zebrafish embryo toxicity. Although the study presents the biological effect of the plant extract in a well-planned manner, the characterization and presentation of the nanoformulated extract has many shortcomings, which makes its manuscript in its current form questionable. I detail my critical comments below:

-       Even in the abstract, it appears that the chemical stability of curcumin is not mentioned as a critical point in terms of PK/PD properties (Abstract:line19-20, Introduction: line 58-60). Unfortunately, this was not further addressed in the authors' work. The real shortcoming of the work is that the known chemical decomposition of curcumin as a result of the nanoformulation is not shown at all, even though this formulation strategy would expect a protective effect. This should definitely be replaced in order to accept the announcement.

-       The most critical point of the announcement is the HPLC-based characterization of curcumin and Curcuma longa(CL) (2.3 and 3.1 respectively, Figure 2). As shown in Fig. 2. it can be seen that the elution of the samples takes place by choosing a relatively long separation procedure. However, curcumin itself is not uniform (Fig2A), which may result from the decomposition of curcumin, and by definition neither is the CL methanolic extract. Basically, this would not be a problem, but the baseline separation of the components does not take place. Accordingly, it is not clear from the presented chromatograms how many components the extract consists of. Furthermore, in the absence of baseline separation, it is not entirely clear how they were able to give the exact curcumin content of the extract. Questions to be clarified: what is the proportion of the decomposition product in curcumin, how many components does the methanol extract contain. Overall, it was not declared whether the measured biological effect could actually come from the decomposition product or from other components of the extract. It would have been advisable to perform the tests using the HPLC-MS method. Why were the chromatograms Fig2A and B not presented at the same wavelength? Did the evaluation also take place at different wavelengths?

-       Another critical point is that no study is presented regarding the release of the active ingredient from the Cl-PCL formulation. In connection with the previous analytical shortcoming, it has not been proven in what proportion and to what extent the components of the extract are released by the nanoformulation. The lack of an answer to this also makes the publication of the manuscript questionable.

-       line 96-97: please provide a reference to the OECD 236 guideline (e.g. web link)

-       line 102-103: The entry of the companies supplying the chemicals is incomplete (name of country, city, possibly branch office!), the names of the Merck and AppliChem companies are listed incorrectly.

-       Line 245-252: for better comprehensibility, it would be important to specify in the text part and not only in the SI, in which dose range the nanoformulated CL extract was tested.

Author Response

Reviewer 2

The manuscript entitled “Protection against paraquat-induced oxidative stress by Curcuma longa extract-loaded polymeric nanoparticles in zebrafish embryos" shows a novel nanoformulation for effective release and effect of curcuma extract for diverse protective effect against paraquat induced zebrafish embryo toxicity. Although the study presents the biological effect of the plant extract in a well-planned manner, the characterization and presentation of the nanoformulated extract has many shortcomings, which makes its manuscript in its current form questionable. I detail my critical comments below:

-       Even in the abstract, it appears that the chemical stability of curcumin is not mentioned as a critical point in terms of PK/PD properties (Abstract:line19-20, Introduction: line 58-60). Unfortunately, this was not further addressed in the authors' work. The real shortcoming of the work is that the known chemical decomposition of curcumin as a result of the nanoformulation is not shown at all, even though this formulation strategy would expect a protective effect. This should definitely be replaced in order to accept the announcement.

We greatly appreciate the comment from the reviewer that helped us improve the present work. In this sense, it was added in the abstract a sentence in line 20 (“Poor chemical stability”), completing the problem in the use of CC as a therapeutic agent. Due to the limitation in the number of words allowed in this section, the pointed subject is approached in more detail in the Introduction section, as follows:

Line 59: “poor chemical stability, low oral bioavailability”

Line 60: “Due to its chemical instability, at physiological pH, CC rapidly degrades to bicyclopentadione by autoxidation. Furthermore, due to the presence of metabolizing enzymes, CC may undergo biotransformation to form reduction products and glucuronide in the liver and other organs [31]. As a result, low serum levels are quantified. The work [32] has shown that after CC oral administration to healthy volunteers, minimal quantities were quantified in serum level (in 10 g only 50.5 ng/mL has quantified). After absorption, CC is subjected to conjugation at various tissue sites, being the liver indicated as the major organ responsible for its metabolism [33]. Additionally, the systemic elimination from the organism is another contributing factor that affects CC bioavailability and consequently interferes with its biological characteristics [33].”

-       The most critical point of the announcement is the HPLC-based characterization of curcumin and Curcuma longa (CL) (2.3 and 3.1 respectively, Figure 2). As shown in Fig. 2. it can be seen that the elution of the samples takes place by choosing a relatively long separation procedure. However, curcumin itself is not uniform (Fig2A), which may result from the decomposition of curcumin, and by definition neither is the CL methanolic extract. Basically, this would not be a problem, but the baseline separation of the components does not take place. Accordingly, it is not clear from the presented chromatograms how many components the extract consists of. Furthermore, in the absence of baseline separation, it is not entirely clear how they were able to give the exact curcumin content of the extract. Questions to be clarified: what is the proportion of the decomposition product in curcumin, how many components does the methanol extract contain. Overall, it was not declared whether the measured biological effect could actually come from the decomposition product or from other components of the extract. It would have been advisable to perform the tests using the HPLC-MS method. Why were the chromatograms Fig2A and B not presented at the same wavelength? Did the evaluation also take place at different wavelengths?

A

B

We thank the comments made by the reviewer that helped us to improve the present work. We clarified the HPLC-DAD analysis and the identification of the compounds (curcumin and curcuminoids) (new Figure 2). Also, we corrected the chromatograms for the same wavelength. We used a long time HPLC analysis to certify the possible presence of other compounds besides the ones mentioned and identified. Using the DAD data, it is easy to certify that no other significant compounds are present, since it gives all the range of spectral data from 230-550 nm; any possible compounds present with different or similar spectra of the curcuminoids would thus be easily detected. There is no detectable degradation of curcumin, but curcumin eluted close to other curcuminoids (which is well known and justifies the difficulty in obtaining commercial pure curcumin). Nevertheless, the separation of the compounds is clear and reached the baseline. The perception of the reviewer was misled because of the full long HPLC runs we showed (Figure 2). But we believe this is also a way to clearly show the inexistence in the extracts of other significant compounds, apart from the ones identified; this can be shown in the additional figures we are sending in supplementary data ( Figure S1). All these topics were addressed in page 3, lines 135-138, page 6, lines 203-211, and in a new Figure 2.

Figure 2: High-performance liquid chromatography (HPLC) analysis of commercial standard curcumin (A) and Curcuma longa (CL) methanolic extract (B). The major compounds present were curcumin (3), demethoxycurcumin (2) and bisdemothoxycurcumin (1).

Figure S1: Detailed analysis of high-performance liquid chromatography (HPLC) analysis of commercial standard curcumin (A) and Curcuma longa (CL) methanolic extract (B) relating to the period between 38 and 44 min, clearly showing the major compounds present were curcumin (3), demethoxycurcumin (2) and bisdemothoxycurcumin (1).

-       Another critical point is that no study is presented regarding the release of the active ingredient from the Cl-PCL formulation. In connection with the previous analytical shortcoming, it has not been proven in what proportion and to what extent the components of the extract are released by the nanoformulation. The lack of an answer to this also makes the publication of the manuscript questionable.

Reference 25 refers to a paper by our research group that presents the optimization of PCL nanoparticles for the delivery of commercially acquired curcumin. In this study, we included a release profile of curcumin from the PCL nanparticles. It reveals that there is sustained release and that, after 168h, approximately 75% of curcumin is released from the nanoparticles. This information has been included in the discussion, from line 378.

-       line 96-97: please provide a reference to the OECD 236 guideline (e.g. web link)

As suggested, the reference has been added, in line 107.

-       Line 102-103: The entry of the companies supplying the chemicals is incomplete (name of country, city, possibly branch office!), the names of the Merck and AppliChem companies are listed incorrectly.

The companies' names were corrected and the country and city of the suppliers were added.

-     Line 245-252: for better comprehensibility, it would be important to specify in the text part and not only in the SI, in which dose range the nanoformulated CL extract was tested.

Tested concentrations of CL extract long exposure were added to the main text (line 259).  

Reviewer 3 Report

The paper by Ana Teixeira et al. directly prepared Curcuma longa-polycaprolactone nanoparticles and confirmed the protective effect against the toxicity of the herbicide PQ by analyzing the antioxidant, cardiac and neuroprotective properties using a zebrafish model.

1. PCL is a biodegradable polymer, and what is the collection efficiency of CL-PCL as a material that collects CL in PCL?

2. In zebrafish model experiments for CL-PCL, intracellular ROS level images should be presented.

3. Authors should better check the manuscript for typos.

4. Provide the CAS number of the substance used.

5 How was Curcuma longa and polycaprolactone dissolved?

6 Add a brief description to all figure legends for the results obtained.

Author Response

Reviewer 3

The paper by Ana Teixeira et al. directly prepared Curcuma longa-polycaprolactone nanoparticles and confirmed the protective effect against the toxicity of the herbicide PQ by analyzing the antioxidant, cardiac and neuroprotective properties using a zebrafish model.

  1. PCL is a biodegradable polymer, and what is the collection efficiency of CL-PCL as a material that collects CL in PCL?

Our reference 25 gives detailed information on optimization of this type of nanoformulation using commercially available curcumin. The selected formulation reaches >70% of encapsulation efficiency. This is now mentioned in the discussion, from line 378.

  1. In zebrafish model experiments for CL-PCL, intracellular ROS level images should be presented.

This data was included as supplementary data (Figures S1 and S2) and discussed in line 389 and line 401.

Figure S2. Reactive oxygen species (ROS) induced by paraquat (PQ) and hydrogen peroxide (H2O2) exposure on zebrafish embryos. At 2 hpf, the embryos were exposed to 0.64 µg/mL of PQ and 1.7 µg/mL of H2O2 for 24 h and, after this incubation time, the mediums were removed and replaced by freshwater. ROS generation were measured 24 h after incubation and 24 h for its removal.

Figure S3. The effects of Curcuma longa extract (CL) exposure on paraquat (PQ)-induced ROS generation in zebrafish embryos. Zebrafish embryos were exposed to PQ for 24 h and treated with CL free, CL-poly-Ɛ-caprolactone (CL-PCL) nanoparticles and gallic acid (GA). The concentrations tested were: 0.64 µg/mL of PQ; 5 µg/mL of CL free, CL-PCL and empty NPs and 25 µg/mL of GA. ROS measurement was performed 24 h after the removal of PQ (48 h).

  1. Authors should better check the manuscript for typos.

Thank you for your suggestion; the document has been revised accordingly.

  1. Provide the CAS number of the substance used.

The CAS number of paraquat was added in line 101 “CAS number 75365-73-0”.

  1. How was Curcuma longa and polycaprolactone dissolved?

The answer to this question is presented in the line 131: “The corresponding mass of PCL and CL were dissolved in 4 mL of acetone”.

  1. Add a brief description to all figure legends for the results obtained.

Thank you for the suggestion. The following information in the figures legend was added or rewritten:

Figure 2. High-performance liquid chromatography (HPLC) analysis of commercial standard curcumin (A) and Curcuma longa (CL) methanolic extract (B). The major compounds present were curcumin (3), demethoxycurcumin (2) and bisdemothoxycurcumin (1).

Figure 3. (…) Loaded nanoparticles present values of mean size and PdI lower than the empty ones and are more negatively charged. Both formulations are stable for at least 4 weeks after preparation. …

Figure 4. (…) CL (free and formulated) reduced the mortality rate. Furthermore, the treatment with CL Free and CL-PCL increased not only the heart frequency of zebrafish embryos exposed to PQ, but also increased the number of embryos with SM and the larvae with FS behavior. …

Round 2

Reviewer 1 Report

The author made good revisions, no comment anymore.